# Mechanisms Affecting Physical Aging and Swelling by Blending an Amphiphilic Component

**DOI:** 10.3390/ijms23042185

**Published:** 2022-02-16

**Authors:** Shifen Huang, Yiming Zhang, Chenhong Wang, Qinghua Xia, Muhammad Saif Ur Rahman, Hao Chen, Charles Han, Ying Liu, Shanshan Xu

**Affiliations:** 1Institute for Advanced Study, Shenzhen University, Shenzhen 518060, China; 2060392007@email.szu.edu.cn (S.H.); fiasanar@szu.edu.cn (M.S.U.R.); mason.hao.chan@gmail.com (H.C.); han.polymer02@yahoo.com (C.H.); 2CAS Key Laboratory for Biomedical Effects of Nanomaterials and Nanosafety & CAS Center for Excellence in Nanoscience, National Center for Nanoscience and Technology of China, Beijing 100190, China; zhangym2021@nanoctr.cn; 3State Key Laboratory of Polymer Physics and Chemistry, Joint Laboratory of Polymer Science and Materials, Beijing National Laboratory for Molecular Sciences, Institute of Chemistry, Chinese Academy of Sciences, Beijing 100190, China; xiaqh@iccas.ac.cn; 4Key Laboratory of Optoelectronic Devices and Systems of Ministry of Education and Guangdong Province, College of Physics and Optoelectronic Engineering, Shenzhen University, Shenzhen 518060, China; 5GBA National Institute for Nanotechnology Innovation, Guangzhou 510700, China

**Keywords:** PLGA, PELA, electrospinning, stability, physical aging, swelling

## Abstract

Polymer blending is a promising method to overcome stability obstacles induced by physical aging and swelling of implant scaffolds prepared from amorphous polymers in biomedical application, since it will not bring potential toxicity compared with chemical modification. However, the mechanism of polymer blending still remains unclearly explained in existing studies that fail to provide theoretical references in material R&D processes for stability improvement of the scaffold during ethylene oxide (EtO) sterilization, long-term storage, and clinical application. In this study, amphiphilic poly(ethylene glycol)-co-poly(lactic acid) (PELA) was blended with amorphous poly(lactic-co-glycolic acid) (PLGA) because of its good miscibility so as to adjust the glass transition temperature (Tg) and hydrophilicity of electrospun PLGA membranes. By characterizing the morphological stability and mechanical performance, the chain movement and the glass transition behavior of the polymer during the physical aging and swelling process were studied. This study revealed the modification mechanism of polymer blending at the molecular chain level, which will contribute to stability improvement and performance adjustment of implant scaffolds in biomedical application.

## 1. Introduction

Polymer materials are increasingly used as medical devices because of their excellent biocompatibility and biodegradability [1,2]. However, for many glassy polymers, physical aging is ubiquitous during their storage or application process. Physical aging will change the structure and properties of the materials [3], seriously affecting the safety and effectiveness of medical devices, and even leading to significant medical accidents. For example, when polyurethane (PU) is used as the insulating layer of a cardiac pacemaker, its aging or degradation will lead to insulation failure and local leakage, causing stimulation and even damage to surrounding muscles [4]. At the same time, pacemaker pacing failure brings life danger to patients. When poly(lactic-co-glycolic acid) (PLGA) is used as absorbable suture, aging may lead to its loss of mechanical properties before wound healing and affect tissue repair [5]. Therefore, it is of great importance to prevent or reduce physical aging and retain the stability of polymer materials during storage and application.

When processed, the polymer is rapidly cooled below its glass transition temperature (Tg) so that the polymer chains are constrained in the high-energy glassy state. Physical aging is a process of transition from thermodynamic non-equilibrium to an equilibrium state, during which excess energy is released through stress relaxation, which is essentially realized by the micro-Brownian motion of molecular chains [6]. In the relaxation process, local chain segments are stacked in parallel under the intermolecular force, forming a new short-range ordered structure [7,8], which reduces the mobility of molecular chains and leads to the decline of material properties [9]. Therefore, the structure and motion of molecular segments play a decisive role in the aging process of polymers. One of the strategies to reduce the physical aging of polymer is improving the flexibility and mobility of molecular chains. In past manufacturing, the method often used was to add a small molecular plasticizer into the polymer materials [10], which could exist between polymer chains to enlarge the intermolecular distance, making the molecular chain segments more flexible and, thus, reducing the Tg of materials. However, the health hazards caused by these compounds have attracted extensive attention in recent years. For example, di-2-ethylhexyl phthalate (DEHP) was extensively added to soft polyvinyl chloride (PVC) medical products to reduce hardness, and its ability to migrate to fatty solutions (such as blood) was proved, which was harmful to the development and reproduction of mammals [11]. Therefore, new solutions are being developed worldwide, mainly including: (1) chemical modification such as polymer grafting [12] or copolymerization [13,14]; (2) physical modification such as polymer blending [15,16]. Compared with chemical modification, physical modification saves more time and money, has lower technical requirements, and is conducive to large-scale production. Through polymer blending, products with stable desired properties can be easily obtained to meet specific clinical needs [17].

In previous studies, we constructed a simple blending system (PLGA/poly(ethylene glycol)-co-poly(lactic acid) (PELA)) to prepare electrospun scaffolds and used them as implantable medical devices such as antiadhesion barriers [18,19], tissue scaffolds [20], and drug delivery systems [21,22,23]. By optimizing the composition and structure, required properties could be obtained such as antiadhesion barriers with a stable morphology and tissue scaffolds with stable mechanical properties and controllable drug--release behavior. However, the mechanism for changing the stability of medical devices through polymer blending has not been clearly explained in existing studies. In addition, how water molecules in the environment affect the swelling or aging process by changing the flexibility of the polymer chains, thus changing the stability and properties of the material, remains unknown. Only by figuring out these mechanisms clearly can we provide a useful theoretical reference for the early design and optimization of implantable biomedical devices to reduce physical aging in the later storage or application of materials so that the devices can perform a consistent and stable function throughout their whole lifecycle. Especially concerning drug-loading systems, drug release is affected by the relaxation behavior of molecular chains [23]. Therefore, a detailed study on the physical aging of polymers below Tg will benefit the design and development of drug-loading systems.

In this study, we continued to use the PLGA/PELA system to prepare electrospun membranes to study the effect of blended PELA on the stability and swelling process of PLGA scaffolds and reveal the inner mechanism. PLGA is a biodegradable polymer approved by the Food and Drug Administration (FDA) and a good model for studying the physical aging process, because polymer chains are physically constrained and confined to a particular scale in the electrospun fibers. Diblock copolymer PELA was blended with PLGA for the following reasons. Firstly, polylactic acid (PLA) segments can enable PELA to be miscible with PLGA, which avoids diffusion of the second component in the swelling process. Secondly, polyethylene glycol (PEG) segments are more flexible and can regulate the Tg of the whole material. Lastly, PEG segments are dissolvable in water and can regulate the hydrophilicity of the whole material. Four groups of PLGA/PELA fibrous membranes (mass ratios of 100/0, 95/5, 90/10, and 85/15) were prepared by electrospinning. The changes in morphological and mechanical properties of fibrous membranes before and after being incubated in the dry and wet states were observed and measured to study the effect of the stability by PELA. Cast films with the dense structure were prepared for comparison and more comprehensive analysis. Glass transition and enthalpy change of the fibrous membranes in dry and wet states, directly indicating the flexibility and relaxation of polymer chains, were characterized by differential scanning calorimetry. The hydration process of the fibrous membrane was characterized by infrared spectrum and water absorption to explore the effect of the hydrophilic component on the swelling process. A reasonable explanation for the improvement of PLGA properties by PELA was given at the molecular chain level and, thus, the mechanism was revealed.

## 2. Results

### 2.1. Stability Obstacles of Pure PLGA Fibrous Membrane during the Whole Lifecycle

Implant medical devices need to be sterilized after preparation and before surgical implantation to reduce the risk of infection. For clinical products, the devices need to ensure the maintenance of intended function within their shelf life, which is generally more than two years. However, both scaffolds in previous studies and our designed electrospun membrane showed difficulty maintaining their fitness of use during the above processes. PLGA is a polymer that can be used in vivo with FDA approval and has been widely used in various medical devices. We prepared a pure PLGA membrane by electrospinning and sterilized it according to the official requirements of ethylene oxide sterilization in the International Organization for Standardization (ISO). It was found that the temperature induced severe physical aging of the material during sterilization, resulting in serious shrinkage of the fiber and change in the macro morphology (Figure 1A). In addition, we also followed the ISO to carry out a shelf-life study of the PLGA electrospun membrane and stored it under recommended conditions. The results showed that the performance of the electrospun membrane changed, and the validity period was shortened due to the atmospheric humidity. As shown in Figure 1B, the fiber began to deform in the third month, strong adhesion between fibers occurred in the sixth month, fibers with integrated morphology were challenging to observe in the twelfth month, and the electrospun membrane finally became transparent. In addition to shelf life, useful life is also essential for products implanted in the body, because it refers to the length of time the material can play a role in the body. However, the mechanical properties of pure PLGA were significantly reduced under the simulated conditions of physiological temperature and physiological environment, and the material became brittle and showed ductile fracture (Figure 1C). If the material became brittle rapidly after implantation, its performance would be significantly affected, and its service life would be significantly shortened.

To sum up, the stability of pure PLGA electrospun membrane will encounter different obstacles in the whole lifecycle. In general, it is affected by temperature/humidity, which induces the physical aging/swelling of the materials. During electrospinning, the solvent volatilizes rapidly under a high voltage electric field so that the polymer chains are rapidly frozen in a high-energy state. When the membrane is stored or used in a thermal environment below its Tg, especially in a hydrothermal environment, the polymer chains gradually relax/swell until they finally reach an energy balance. These processes are related to chain movement in the electrospun fibers, which is determined by the materials’ glass transition/swelling properties. Therefore, we hoped to use a simple physical modification method in this study, namely, polymer blending, to modify the pure PLGA electrospun membrane, change its physical aging/swelling process to improve the stability, and reveal the mechanism (Figure 2A). Here, we selected PELA to be blended with PLGA for the following reasons: PELA has good miscibility with PLGA because one end of PELA is PLA; the Tg of PELA is approximately −41 °C, and it could reduce the Tg of PLGA; PEG segment can improve the hydrophilicity of PLGA and change the water absorption of PLGA membrane.

### 2.2. Preparation of Fibrous Membrane with the Same Physical Structure and Different Chemical Components

Four groups of PLGA/PELA electrospun membranes with different proportions (i.e., 100/0, 95/5, 90/10, and 85/15) were prepared by electrospinning. In order to control a single variable and only study the influence of components on properties, we adjusted the spinning parameters so that the four groups of fibrous membranes had the same physical structure but a different chemical component. As shown in Figure 2B, the four groups of electrospun membranes had a highly consistent three-dimensional network structure; the fibers were smooth and solid, and the fiber diameter distribution mainly ranged from 0.8 to 2.4 μm. The addition of PELA did not change the morphology of the fibers, which to some extent proved that PELA had good miscibility with PLGA. Because the molecular weight of PELA was small and its Tg was far less than room temperature, the addition of PELA would change the Tg of PLGA and affect its molecular chain movement. With the increase of PELA content from 0% to 15%, the infiltration time of water on the membranes decreased rapidly from approximately 60 to 2 min, indicating that the addition of PELA could significantly enhance the hydrophilicity of the PLGA electrospun membrane. Therefore, we could reveal how PELA affects the physical aging/swelling process by changing the Tg/hydrophilicity of pure PLGA membrane in the following study.

### 2.3. Morphological Stability and Mechanical Performance of Fibrous Membrane

Stability is crucial for a biomedical implant scaffold when designed to work inside the body for an extended period. Maintaining morphological integrity and stability and mechanical strength and toughness is a crucial necessity for the scaffolds to perform a stable and long-lasting function in vivo. However, physical aging, which impairs stability severely, is inevitable in the whole lifecycle of the polymer scaffolds, especially PLGA fibrous membrane. During the electrospinning process, the rapid volatilization of solvent under a high voltage field quickly froze the polymer chains in a high-energy state. When the membrane was stored or used at a temperature lower than its Tg, the polymer chains relaxed gradually until reaching an energetic equilibrium state, considered physical aging. Many researchers have already pointed out that physiological temperature would induce severe physical aging of PLGA [24].

To study the effect of PLA-b-PEG on the stability of PLGA-based membranes, we incubated the membranes in phosphate buffered saline (PBS) at 37 °C for two days to simulate the physiological environment. Not surprisingly, 100/0 membrane (pure PLGA) showed severe membrane shrinkage, whereas 85/15 membrane retained better morphological stability as shown in Figure 3A,B. In other words, the addition of PLA-b-PEG increased the morphological stability of the PLGA fibrous membrane during physical aging. The shrinkage of the PLGA membrane was mainly attributed to the relaxation behavior of polymer chains in the fibers. When the temperature increased, the thermal motion of molecular chains in electrospun fibers became violent, and they gradually relaxed from a rigid state to a random coil shape, causing the fibers to deform. The relaxation behavior of the molecular chains was mainly related to Tg and the thermal history of the material.

In addition to morphological stability, mechanical properties are also important for the implanted scaffold. The embrittlement of electrospun membrane during storage is difficult to judge directly with the naked eye. If inadvertently implanted into the body, the performance of the scaffold would be damaged, and the life safety of patients would be threatened. Therefore, dry and wet states were used as comparative experiments here. Dry and wet membranes (incubated in PBS) were incubated in the incubator at 37 °C, and the tensile strength, modulus, and elongation at break were tested at different incubation time spots (i.e., 0, 2, 5, and 12 h).

As shown in Figure 3C, the elongation at break decreased significantly with time, since the polymer physically aged under the influence of temperature. The micro-Brownian motion of the molecular chains made some chain segments form a local parallel arrangement under the intermolecular force. This local ordered structure could be considered a crystalline region, and the number of crystals increased with the extension of aging time. Elongation at break microscopically reflected the relative displacement between molecular chains in the tensile process. The ordered structure between these molecular chains made it difficult for them to slide relatively during the stretching process of the membrane, so the elongation at break decreased with time.

Most importantly, we found that the elongation at break of the 85/15 membrane was significantly greater than that of the 100/0 membrane, indicating that the addition of PELA improved the toughness of the PLGA electrospun membrane, which was mainly related to their different Tg. When both were at the same ambient temperature, the higher the Tg, the worse the mobility of the molecular chains, relatively increasing the difficulty of slide; therefore, the toughness was worse.

With the increase in incubation time, the tensile strength and modulus of the two groups of fibrous membranes were almost unchanged in both the dry and wet states. For polymer networks, these two mechanical properties could be considered to reflect the force that the chain segments were able to bear during stretching, which was mainly related to the number of entanglements formed between molecular chains. It is speculated that the entanglement was not fully opened when tested at the time spots during the experiment, so both tensile strength and modulus were not affected.

In order to further verify the mechanism of PELA improving PLGA’s performance, we also prepared cast films with a dense structure, and the proportions of PLGA/PELA were 100/0 and 85/15, respectively. Compared with the 100/0 and 85/15 electrospun membranes, the material with the same polymer ratio would have the same Tg. However, different from the electrospinning process, the preparation of cast film was a slow quenching process during which polymer chains were subject to continuous stress relaxation, resulting in easier and faster local crystallization of polymer chains in the cast film during physical aging, which was explicitly manifested in the rapid decrease in elongation at break (as shown in Figure 3C). Therefore, we could conclude that the mechanism of PELA improving the stability of PLGA electrospun membrane is that PELA affected the stress relaxation of molecular chains in the electrospun fibers by adjusting material Tg to change the chain movement. It was also noted that the fibrous membrane had lower tensile strength and modulus compared with the cast film. Therefore, the electrospun membrane would have better compliance than the cast membrane when used in vivo, which is significant for clinical application.

### 2.4. Glass Transition and Enthalpy Change of Fibrous Membrane

According to the previous experimental results, we know that PELA can significantly improve the morphological stability and mechanical properties of PLGA electrospun membranes. Analysis showed that this improvement was based on Tg regulation that affected chain segments’ stress relaxation and movement. For example, when the temperature was closer to the Tg of the material, the molecular chain activity increased and showed better toughness. In the process of glass transition, fibers with more enthalpy relaxation would have more significant deformation. In order to verify our analysis, we wanted to measure the Tg and enthalpy of the material. Differential scanning calorimetry (DSC) could measure the Tg of the material and obtain the enthalpy relaxation in the process of glass transition. According to DSC, in the heating process, when the temperature was still lower than the Tg of the fiber, the chain segments were frozen in a glassy state. At this time, the motion of the long polymer chains was bound and could only vibrate limitedly in situ. When the temperature gradually rose to approximately the Tg of the fiber, the chain motion became active, and the excess energy was released in the form of enthalpy in the process of chain relaxation to realize the transition from a non-equilibrium to an equilibrium state. Therefore, an endothermic peak could be observed near Tg in the DSC curve.

According to the method in Figure 4A, four groups of fibrous membranes with different proportions were tested in the dry and wet states, respectively. The results showed only one Tg peak in all fibrous membranes, whether in a dry or wet state, which indicates that PLGA and PELA had good miscibility and could form fibers with uniform texture after high-voltage electrospinning. With the increase in PELA content, the Tg of the fibrous membranes gradually decreased (Figure 4B). Since PELA had low Tg, it is not difficult to explain this result according to the principle of polymer blending. Among them, we found that the Tg of the 85/15 membrane decreased from 49 (pure PLGA membrane) to 36 °C, so it could be considered to be in a rubbery state when incubated at 37 °C, which also explains why its toughness was significantly better than that of the 100/0 membrane in the tensile experiment.

As can be seen from the DSC curve (Figure 4C), the 100/0 membrane had a pronounced endothermic peak in the dry state, and the peak area decreased with the increase in PELA content. The endothermic peak reflected the excess energy released by the molecular chain during the relaxation process, which was associated with the Tg and quenching history of the material. For the 100/0 membrane with a Tg of 49 °C, its molecular chain was rapidly stretched and cooled to room temperature (approximately 25 °C) during spinning. At this time, the polymer chains were in an unstable high-energy state. During the heating process of DSC, when the temperature was raised to near Tg, the internal stress of the molecular chains was relaxed through the movement of the chain segments so that an apparent endothermic peak could be observed on the DSC curve. With the increase in PELA content, the Tg of the fibrous membrane decreased. The 85/15 membrane had a Tg of 36 °C, which was closer to room temperature than the 100/0 membrane. Although the polymer chains were also frozen at room temperature, the Brownian motion was more active than the 100/0 membrane. During the quenching process, some molecular chains of the 85/15 membrane released part of the internal stress through segment motion, so the endothermic peak area on its DSC curve was smaller.

However, it can be observed that the endothermic peaks were not evident in the wet state. Water molecules acted as plasticizers, increasing the free volume between polymers thus increasing the activity of molecular chains and reducing the Tg of fibrous membranes in the wet state (Figure 4B) [25]. It was speculated that the molecular chains released the internal stress in advance in a wet state so that no apparent endothermic peak could be observed on the DSC curve when experimented. For electrospun membranes, when the polymer chains inside the fibers relaxed from the non-equilibrium stretching shape to the equilibrium random coil shape, the fiber would shrink, which would lead to the shrinkage of the fibrous membrane macroscopically. In this process, if the molecular chain relaxed more, the deformation would be more significant and the fibrous membrane would shrink more seriously. Therefore, it was observed in the previous results that the 100/0 membrane shrank significantly after immersion in PBS for two days, while the structure of the 85/15 membrane was more stable.

### 2.5. Swelling Behavior of the Fibrous Membrane

For implant devices, the stability should be maintained in a dry state (such as storage or transportation) and in a wet state (such as an in vivo application). From the mechanical properties results, we found no significant difference in the process of toughness deterioration of the 100/0 membrane in the dry and wet states, while that of the 85/15 membrane in the wet state was significantly faster than that in a dry state. However, PELA has been proved to improve the morphological stability and mechanical properties of the PLGA membrane. Nevertheless, it was found that the addition of PELA worsened the mechanical properties of the 85/15 membrane in the wet state. Therefore, we characterized the swelling behavior of the 100/0 membrane and the 85/15 membrane to explore the role of water molecules in the swelling process of different component fibers.

The micro morphology of electrospun membranes after swelling (PBS, 37 °C) for two days was characterized. As shown in Figure 5A, all of the fibers became curly after swelling, and the degree of curl decreased with increasing PELA content, resulting from the enthalpy relaxation of molecular chains inside the fibers, which had already been analyzed earlier. In addition, the fiber diameters became larger after swelling due to the increase in the distance between the polymer chains with the existence of water molecules. Theoretically, if more water entered the polymer (for example, when the membrane was more hydrophilic), the fiber diameter would increase more. However, the phenomenon observed here did not correspond. It can be seen from the comparative data of fiber diameters in Figure 5B that the fiber diameter of the 85/15 membrane during swelling was always less than that of the 100/0 membrane.

A previous study found that the fibers of the 100/0 membrane were smooth and solid after swelling for two days, while small cavities appeared on the fibers of the 85/15 membrane [18]. This could be attributed to the phase separation of the blended polymer during swelling, namely, PELA and PLGA. Because PLA segments had better compatibility with PLGA, PELA would not diffuse from the system to the surrounding solvent during swelling. Because the hydrophilicity of the PEG segments was better, it might exist at the end close to the solvent after phase separation.

The premise of swelling was hydration. For electrospun fibers, hydration occurred successively from the outside to the inside. The hydration process was affected by the interaction between polymer and water molecules. As shown in Figure 5C,D, the interaction between PLGA/PELA and water molecules during swelling was characterized by FTIR. It could be seen from the infrared spectrum that there were two broad peaks at 3400 and 2400 cm^−1^, respectively. According to the structure of PLGA and PELA in Figure 2A and the detection principle of the infrared spectrum, it could be inferred that the 3400 cm^−1^ peak was the hydrogen bond between PLGA or PLA and water molecules, while the 2400 cm^−1^ peak was the hydrogen bond between PEG and water molecules, and it could be considered that the hydrogen bond formed by PEG was more substantial than that formed by PLGA or PLA. With the extension of swelling time, the absorption value of the 2400 cm^−1^ peak increased, indicating that more hydrogen bonds were formed between PEG and water molecules.

Based on the above analysis, we can deduce that the swelling process of fibers in the 85/15 membrane was as follows: When the fibers were immersed in water, water molecules gradually entered the fibers. In this process, the phases of PELA and PLGA gradually separated, and cavities formed on the fiber with the PEG segments faced outward and the PLA segments faced inward. Strong hydrogen interactions formed between the PEG segments and water molecules that “locked” a large number of water molecules around PEG and hindered their entrance into the PLGA chains. Although the 85/15 membrane showed a high water absorption rate and reached absorption saturation in less than 10 h (Figure 5E), most of the water existed in the gap between the fibers or the cavities formed by phase separation in the fibers. For the 100/0 membrane, without the water retention effect of PEG, water molecules could continuously enter into the fibers, increasing the distance between polymer chains and, thus, increasing the fiber diameter. Therefore, the increase in the fiber diameter of the 100/0 membrane during swelling was more significant than that of the 85/15 membrane.

Generally speaking, water molecules can be regarded as plasticizers because they increase the mobility of molecular chains. However, we found that the toughness of the 85/15 membrane in the wet state was lower than that in the dry state. According to Figure 4B, the Tg of the 85/15 membrane in the wet state (approximately 39 °C) was more significant than that in the dry state (approximately 36 °C). It was thought that the strong interaction between PEG and water molecules reduced the mobility of the molecular chains. Theoretically, the toughness of the 100/0 membrane in the wet state should be greater than that in the dry state, but we did not observe a significant difference from the results in Figure 3C. This might be because the 100/0 membrane was not entirely hydrated at the time spots of the mechanical properties test, where the water absorption was only approximately 40% after 12 h.

## 3. Discussion

In this research, we studied the effect and mechanism of the second component PELA on reducing the physical aging of PLGA. PELA could significantly improve the morphological stability and mechanical properties of the PLGA membrane (Figure 6). With increasing PELA content, the Tg and enthalpy relaxation of the electrospun membranes decreased gradually. After the same quenching process and being kept in the same environment, the fiber with low Tg released part of the internal stress into the environment during quenching, and the enthalpy released during aging was less; thus, the fiber deformation and the membrane shrinkage was relatively slight. The lower the Tg, the higher the mobility of the polymer chains, so the mechanical properties were better. In addition, we also used the amphiphilic of PELA to explore the effect of water molecules on the swelling process of fibers when the content of PELA increased to a certain amount due to the strong hydrogen interaction formed between PEG segments and water molecules and the water retention ability of PEG. The activities of chain movement were constrained, which reduced the swelling of fibers and made the toughness relatively poor.

The findings in this study could offer numerous theoretical references for various scientific fields. Firstly, the production, storage, and application of polymeric electrospun scaffolds could be guided. For example, during ETO sterilization, the high temperature would cause membrane shrinkage. The enthalpy relaxation of the scaffold during aging could be reduced by adding a second component with lower Tg to reduce the deformation caused by stress relaxation. In order to reduce aging, the material should be kept at a low temperature as far as possible to reduce the thermal motion of molecular chains. During long-term storage, the environmental humidity should be reduced as much as possible to avoid swelling caused by water molecules penetrating the PLGA membrane, or hydrophilic components could be added to reduce the penetration of water molecules by holding water molecules in the hydrophilic part. To create material with better mechanical properties, a second component with lower Tg could be added to enhance the toughness by improving the activity of the molecular chains. However, it should be noted here that if the low Tg component was hydrophilic at the same time, when the content was too much, the material might show worse toughness in the wet state than in the dry state. Therefore, the content should be controlled within a specific range, rather than the more the better.

Secondly, by exploring the relaxation and swelling behavior of polymers, we developed a deeper understanding of the drug-release behavior of drug-loading systems with different hydrophilicity. In a study by Wang et al., they prepared the same proportion of PLGA/PELA drug-loaded electrospun membranes (i.e., 100/0, 95/5, 90/10, and 85/15) and found that the drug had a significant burst release with the increase in PELA content while showing sustained release in the pure PLGA membrane [18]. Wu et al. revealed the evolution of drugs released from electrospun membranes to surrounding medium [23]: (1) water penetrated the fiber, and the drug aggregation began to expand in the fiber; (2) the drug began to dissolve and gradually evenly distributed in the fiber; (3) the drug excreted the fiber and diffused into the surrounding medium. In general, the drug needed to travel first within the fibers and then between the fibers. Another study by Wu et al. revealed that drug release in electrospun membranes could be divided into three stages [22], and they discussed the influence factors of each stage: (1) stage I was mainly affected by the hydrophilicity and Tg of the polymer chain; (2) stage II was mainly affected by fiber swelling; (3) stage III was mainly affected by polymer degradation.

Combined with the results of this study, we offer a deeper understanding of drug-release behavior in drug-loaded membranes with different hydrophilicities as follows: For pure PLGA membrane, when immersed in a solution, water molecules begin to enter the fiber. Because the diffusion of drugs inside the fiber is mainly affected by the relaxation of the molecular chains and the large Tg of pure PLGA indicates lower mobility, the diffusion of drugs into PLGA fiber is slow, resulting in only a tiny amount of drugs being transported into the surrounding medium. Therefore, PLGA has a slower initial drug-release rate and lower initial release amount. When more and more water molecules enter, the fiber begins to seriously swell, leading to fiber fusion. The membrane becomes a gel-like structure with the travel channels of the drugs between the fibers blocked. The drug can only diffuse from the upper and bottom of the swollen membrane. Therefore, the drug-release rate is slowed down, and it takes a long time (until the polymer is degraded) to completely release the drug.

For the 85/15 membrane, the existence of PELA reduced the Tg of the overall material, resulting in enhanced molecular chain mobility. PELA also enhanced the hydrophilicity of the fiber, and more water entered the fiber to contact with the drug at the initial stage compared to PLGA, which promoted the dissolution of the drug and made the drug spread out of the fiber faster. Therefore, the 85/15 membrane showed a sizeable initial release rate and a large initial release amount. However, in the second stage, the hydrogen bond restricted the migration of water molecules, which slowed down the swelling of the fibers, so the fibers did not fuse like the pure PLGA fibers (Figure 6D). The channels of drug diffusion from the fibers to the surrounding medium were not blocked. Therefore, the drug still maintained a faster release rate than in the PLGA membrane. As we mentioned earlier, only when PELA reached a certain content would the hydrogen interaction play a leading role in chain mobility and fiber swelling. Zhang et al. found that in PLGA/PELA (93/7) membrane, PELA could not prevent the fibers from fusing [21]. Understanding the drug-release behavior of electrospun membranes with different Tg and hydrophilicities would be conducive to the design and optimization of the drug-loading system to develop drug-loaded membranes with specific drug–release curves according to different clinical conditions needs.

Finally, this glass transition property was also significant for shape memory polymers (SMPs) [26,27]. SMPs can be deformed into a temporary shape under external force at a temperature lower than Tg and then re-quenched into a permanent shape in the design state under external force at a temperature higher than Tg. As an intelligent material, SMP is expected to be widely used in medical devices [28] such as intelligent sutures [29] and intelligent drug release [30]. Its unique shape memory function will simplify complicated operations during surgical treatment, significantly reduce the pain and risk faced by patients, and bring convenience to doctors. Research and exploration of SMPs have good social significance and good medical prospects. If the characteristics and mechanism of glass transition of materials in different environments can be explained clearly, it will help control SMPs better and improve their function.

## 4. Materials and Methods

### 4.1. Materials

Two copolymers, PLGA (Mw = 60 k, LA/GA = 75/25) and PLA-b-PEG (Mn = 10 k, LA/EG = 50/50, mass ratio), were purchased from Ji’nan Daigang Biology Engineer Co., LTD. N, N-dimethyl formamide (DMF) and acetone double-distilled water were obtained from Beijing Chem. Co. (Beijing, China). Ultrapure water (18.2 MΩ∙cm) was obtained from a Millipore Simplicity unit. Deuterium oxide (D, 99.9%, Cambridge Isotope Laboratories, Inc., Middlesex County, NJ, USA) was used for water diffusion studies. All reagents were used as received.

### 4.2. Fabrication of Electrospun Membrane

Polymer solutions consisting of PLGA or PLGA/PLA-b-PEG were prepared in a mixed solvent of N, N-dimethyl formamide (DMF) and acetone (V_DMF_/V_Acetone_ = 5/5). The total concentration of the solution was 50 w/v% (w in g and v in mL). The weight ratio of PLGA 60 k to PLA-b-PEG 10 k was 100/0, 95/5, 90/10, and 85/15. The electrospinning process was carried out in a sterile environment at 20 kV with a steady flow rate of 10 μL/min (spinneret with a hole diameter of 0.3 mm). The electrospun fibers were collected on a metal drum (as an electrode, 9 cm in diameter, and a TCD (the tip-to-collector distance) of 15 cm) rotating at approximately 120 rpm. The resulting scaffolds contained interconnected webs of fibers 1~3 microns in diameter, which were vacuum dried at room temperature for 3 days prior to usage.

Post-processing, the scaffolds underwent a hot press of 150 g/cm^2^ at 50 °C for 20 min in a sheet mold before usage. This process helped reduce the in vivo shrinkage of the PLGA-based electrospun scaffold by reducing the stress relaxation of the fibers.

### 4.3. Sterilization Process

The prepared electrospun membranes were sterilized with ethylene oxide (EtO) following the official requirement of ISO 11135 with verified manufacturer parameters. The macro morphology of the membranes was photographed directly. The micro morphology of the fibers was characterized by scanning electron microscope (SEM, JEOL JSM-6700F, Tokyo, Japan).

### 4.4. Shelf-Life Study

The prepared electrospun membranes were stored in a constant environment with temperature of 25 ± 5 °C and relative humidity of 60 ± 20%. All the conditions followed the official requirement of ISO 11979. The macro morphology of the membranes was photographed directly. The micro morphology of the fibers was characterized by scanning electron microscope (SEM, JEOL JSM-6700F, Japan).

### 4.5. Characterizations

The morphologies of the electrospun membranes were observed using a scanning electron microscope (SEM, JEOL JSM-6700F, Japan) at an accelerating voltage of 5 kV. The swelled membranes were vacuum dried at room temperature before characterization. Each sample was sputter-coated with platinum for analysis.

The water contact angle was measured using a sessile drop method using a Digital Contact Angle Measurement System with a CCD camera (JC2000A). A water droplet of 5 μL was used, and the image was frozen to measure the static contact angle. The infiltration time of water droplets was measured during the contact angle test, while a water droplet of 200 μL was used.

After 2 days of in vitro incubation, the scaffolds were evaluated for a stress–strain response with a stretch velocity of 10 mm/min using a tensile testing machine (Series IX Automated Materials Testing System, Instron Co., Boston, MA, USA). Scaffolds were cut in a rectangle shape with an area of 10 × 50 mm^2^ prior to the tensile test. The thickness of each membrane was measured using a micrometer, and the reported tensile properties were the average obtained from five specimens. The elongation at break (%) and elastic modulus (MPa) of membranes was calculated from the stress–strain curves.

Time-resolved ATR-FTIR spectroscopy was carried out, and ATR-FTIR spectra were recorded on an infrared spectrometer (IR, BRUKER TENSOR 27, Germany) equipped with a DTGS detector and an ATR accessory (Golden Gate ATR, Specac GS10642). All spectra were collected at a 4 cm^−1^ resolution with 32 scans in the range of 4000–500 cm^−1^.

The thermal properties were investigated using a differential scanning calorimeter (DSC, TA Q2000, NewCastle, DE, USA) under a nitrogen atmosphere with a heating rate of 20 °C/min.

## 5. Conclusions

In this work, the effect and mechanism of PELA on improving PLGA membranes were studied in detail. Compared with pure PLGA membrane, PLGA/PELA membrane containing 15% PELA could significantly reduce the damage of physical aging to the structure and properties, which represented better morphological stability, better mechanical properties, and slower swelling behavior. The primary modification mechanism was to reduce Tg by PELA to improve chain mobility and adjust hydrophilicity to control the migration of water molecules. This work revealed the modification mechanism of polymer blending, showing the feasibility of physically modifying the materials through polymer blending, providing a theoretical reference for the design and optimization of blended materials. This is especially conducive to the development of medical devices.

## Figures and Tables

**Figure 1 ijms-23-02185-f001:**
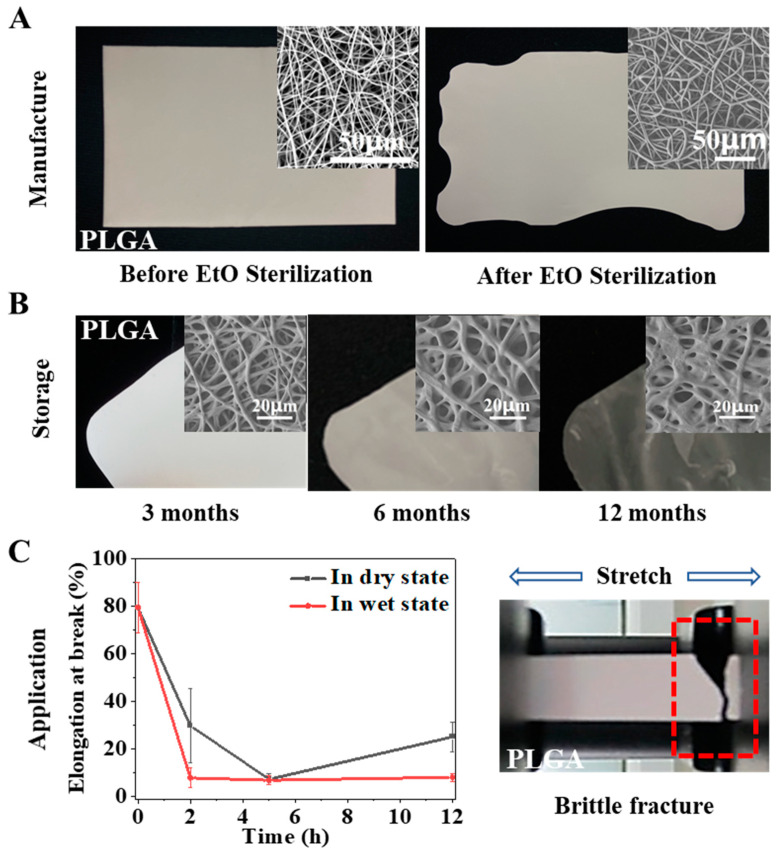
Stability obstacles of PLGA membrane. Morphology and performance changes of pure PLGA electrospun membrane during manufacture (**A**); storage (**B**); application (**C**).

**Figure 2 ijms-23-02185-f002:**
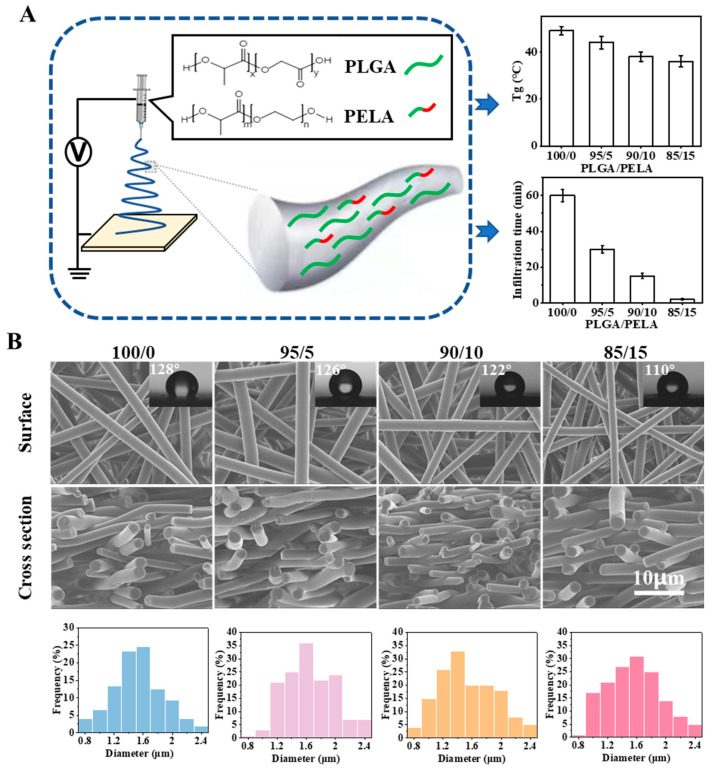
Preparation of fibrous membranes with the same physical structure and different chemical components. (**A**) Schematic illustration of blending PLGA with PELA; (**B**) SEM images, contact angle images, and diameter distributions of the electrospun membranes (i.e., 100/0, 95/5, 90/10, and 85/15). Here, 100/0, 95/5, 90/10, and 85/15 indicate the weight ratio of PLGA to PELA in the membranes.

**Figure 3 ijms-23-02185-f003:**
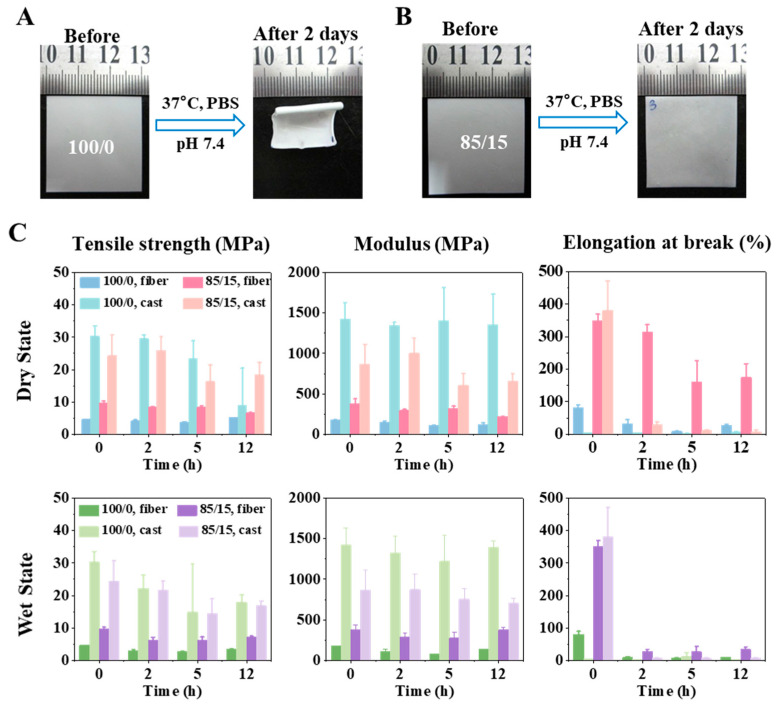
Morphological stability and mechanical performance of fibrous membrane. The macro morphological change before and after swelling in PBS of 100/0 (**A**) and 85/15 (**B**) electrospun membranes. (**C**) Tensile strength (**left**), modulus (**middle**), and elongation at break (**right**) of 100/0 and 85/15 membranes (fibrous membranes and cast films) in a dry state (**top**) and wet state (**bottom**).

**Figure 4 ijms-23-02185-f004:**
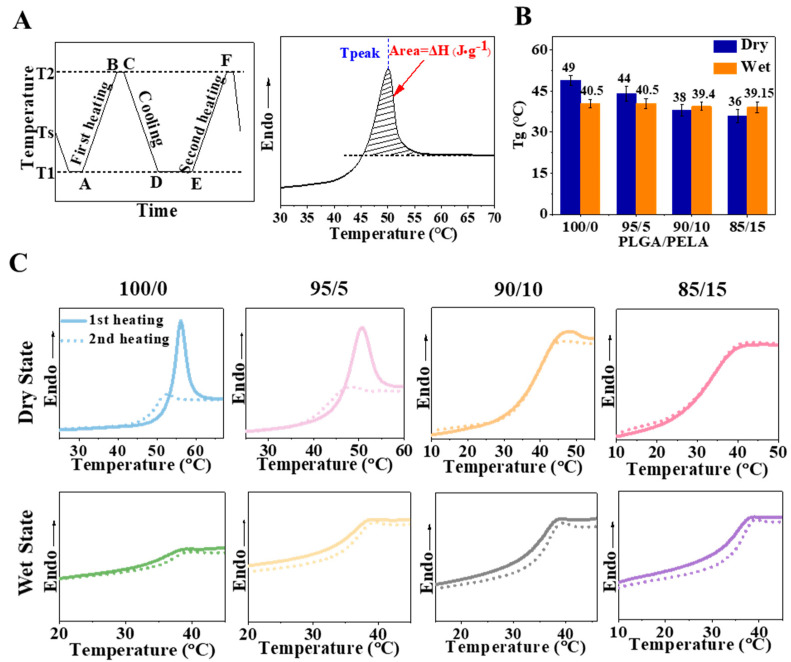
Glass transition and enthalpy change of fibrous membrane. (**A**) Program of DSC process (**left**) and determination of peak temperature and enthalpy change from DSC curve (**right**). T1, Ts, and T2 were 0, 40 (standby temperature), and 80 °C, respectively. (**B**) The glass transition temperature of the fibrous membrane in the dry and wet state. (**C**) DSC curve of the fibrous membrane in the dry state (**top**) and wet state (**bottom**).

**Figure 5 ijms-23-02185-f005:**
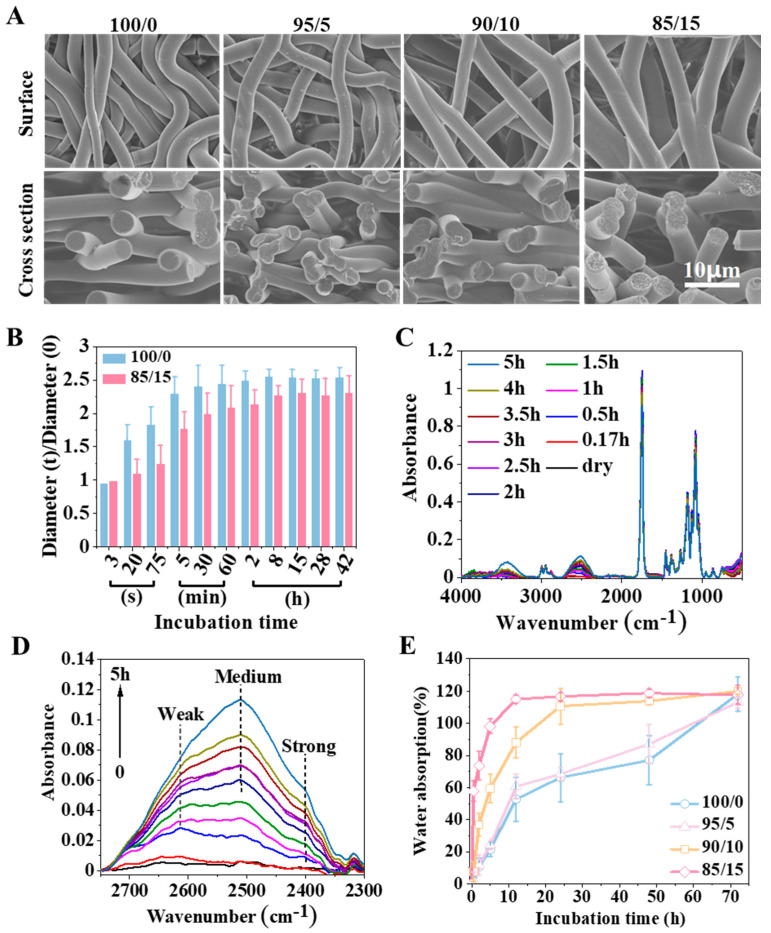
Swelling behavior of fibrous membrane. (**A**) SEM images of the electrospun membranes (i.e., 100/0, 95/5, 90/10, and 85/15). (**B**) Diameter changes of the 100/0 and 85/15 electrospun fibers. Here, 0 and t represent the initial state and the state at time t, respectively. (**C**,**D**) ATR-FTIR spectra for the process of deuterated water diffusion in the 85/15 membranes. (**E**) Water absorption of the electrospun membranes (i.e., 100/0, 95/5, 90/10, and 85/15) during swelling.

**Figure 6 ijms-23-02185-f006:**
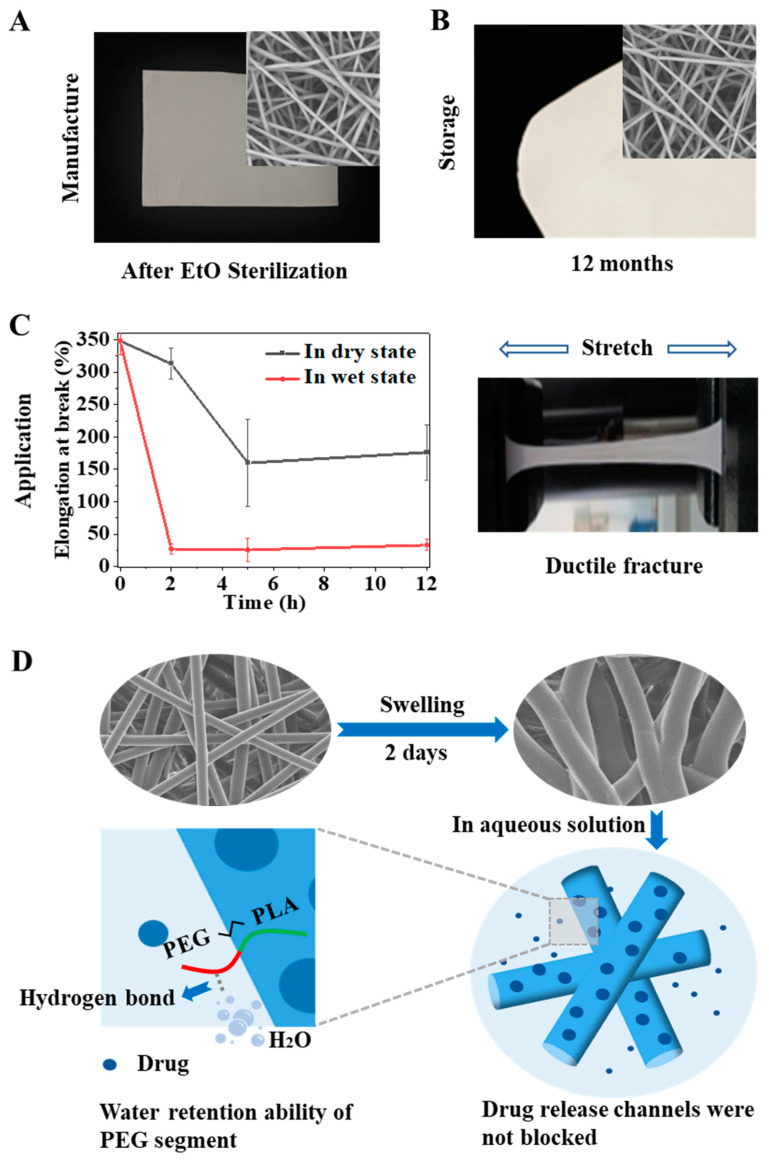
Stability improvement of the PLGA/PELA 85/15 membrane. Improvement in the morphology and performance of the PLGA/PELA 85/15 electrospun membranes during manufacture (**A**); storage (**B**); application (**C**). (**D**) Schematic illustration of the drug-release behavior of the 85/15 membrane in aqueous solution.

## Data Availability

Not applicable.

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
