# Peer review of "Mechanisms Affecting Physical Aging and Swelling by Blending an Amphiphilic Component"

_ijms, 2022, doi:10.3390/ijms23042185_

Round 1
Reviewer 1 Report
PLGA fibrous membranes are attended materials for biomedical applications. The authors investigated the effect of PELA addition in PLGA fibrous membrane produced by electrospinning process on the physical properties and aging. The results would be useful to improve the properties of the PLGA electrospun membrane. Also, the authors had well applied the results to understand the function of biomaterials reported previously. Those are supported by the experimental results. The manuscript is well written and I recommend it for acceptance after some minor points are addressed.
· Figure 5 A and Experimental section: The more detailed description of SEM sample preparation would be needed. I think that sample morphology would be changed by the drying process. Are those freeze-dried samples?
· Line 280-281: It would be an easily deducible result that the Tg of PLGA/PELA system was decreased by adding PELA with the lower Tg than that of PLGA. Why did the authors think that results in Figure 4 B were difficult to explain. Please describe or add a related reference on the principle of polymer blending which the authors mentioned.
· Please check the reference list. I found some mismatches between reference No. and the description in the manuscript.
Author Response
PLGA fibrous membranes are attended materials for biomedical applications. The authors investigated the effect of PELA addition in PLGA fibrous membrane produced by electrospinning process on the physical properties and aging. The results would be useful to improve the properties of the PLGA electrospun membrane. Also, the authors had well applied the results to understand the function of biomaterials reported previously. Those are supported by the experimental results. The manuscript is well written and I recommend it for acceptance after some minor points are addressed.
- Figure 5 A and Experimental section: The more detailed description of SEM sample preparation would be needed. I think that sample morphology would be changed by the drying process. Are those freeze-dried samples?
Reply: Thanks a lot for the reviewer’s professional suggestions. The swelled membranes were vacuum dried at room temperature before characterization. And the change of morphology during the drying process could be ignored since the Tg of the smple was higher than room temperature.
- Line 280-281: It would be an easily deducible result that the Tg of PLGA/PELA system was decreased by adding PELA with the lower Tg than that of PLGA. Why did the authors think that results in Figure 4 B were difficult to explain. Please describe or add a related reference on the principle of polymer blending which the authors mentioned.
Reply: Thanks a lot for the reviewer’s professional suggestions. It is easy to explain why the Tg of PLGA/PELA system was decreased by adding PELA. According to the principle, if the two polymer components are miscible, the blend is a homogeneous system with only one glass transition temperature which is between the Tg of the two polymers. The conclusion has been mentioned clearly in the manuscript.
- Please check the reference list. I found some mismatches between reference No. and the description in the manuscript.
Reply: Thanks a lot for the reviewer’s careful check. We have revised the reference number in the revised manuscript.

Reviewer 2 Report
Huang’s et al. manuscript is devoted to the preparation of biodegradable composite materials via electrospinning technique for biomedical applications. I can indicate that manuscript is well structured, written and discussed. The materials of different composition, treated and non-treated with EtO, were properly characterised with a number of physicochemical methods such as SEM, DSC, wettability, swelling and mechanical tests. The speculation on the mechanism of the reduction of the physical ageing of the composites containing PELA was proposed. In my opinion, this manuscript can be accepted after minor revision.
1) Please provide scale bars to SEM images in Figure 1.
2) Among others, Figure 2 illustrates the results of contact angles measurements. However, I did not find the certain values of those angles in the text or in the figure. I think it would be useful information to track the changes in this property.
3) Line 196. It seems to be better replace term “cultured” with “incubated”.
Author Response
Huang’s et al. manuscript is devoted to the preparation of biodegradable composite materials via electrospinning technique for biomedical applications. I can indicate that manuscript is well structured, written and discussed. The materials of different composition, treated and non-treated with EtO, were properly characterised with a number of physicochemical methods such as SEM, DSC, wettability, swelling and mechanical tests. The speculation on the mechanism of the reduction of the physical ageing of the composites containing PELA was proposed. In my opinion, this manuscript can be accepted after minor revision.
- Please provide scale bars to SEM images in Figure 1.
Reply: Thanks a lot for the reviewer’s careful check. The scale bars have been added to the SEM images in Figure 1.
- Among others, Figure 2 illustrates the results of contact angles measurements. However, I did not find the certain values of those angles in the text or in the figure. I think it would be useful information to track the changes in this property.
Reply: Thanks a lot for the reviewer’s careful check. The values of the contact angles were added in Figure 2.
- Line 196. It seems to be better replace term “cultured” with “incubated”.
Reply: Thanks a lot for the reviewer’s professional suggestions. The “cultured” has been replaced with “incubated” in the revised manuscript.
